# Extracorporeal Cardiopulmonary Resuscitation: A Narrative Review and Establishment of a Sustainable Program

**DOI:** 10.3390/medicina58121815

**Published:** 2022-12-09

**Authors:** Chris M. Cassara, Micah T. Long, John T. Dollerschell, Floria Chae, David J. Hall, Gozde Demiralp, Matthew J. Stampfl, Brittney Bernardoni, Daniel P. McCarthy, Joshua M. Glazer

**Affiliations:** 1Department of Anesthesiology, University of Wisconsin Hospitals & Clinics, 600 Highland Ave., Madison, WI 53792, USA; 2Department of Anesthesiology, Ohio State University Wexner Medical Center, 370 W. 9th Ave., Columbus, OH 43210, USA; 3Department of Surgery, University of Wisconsin Hospitals & Clinics, 600 Highland Ave., Madison, WI 53792, USA; 4Department of Emergency Medicine, University of Wisconsin Hospitals & Clinics, 600 Highland Ave., Madison, WI 53792, USA

**Keywords:** extracorporeal, cardiopulmonary resuscitation, post-cannulation management, program development, ECPR vs. CCPR

## Abstract

The rates of survival with functional recovery for out of hospital cardiac arrest remain unacceptably low. Extracorporeal cardiopulmonary resuscitation (ECPR) quickly resolves the low-flow state of conventional cardiopulmonary resuscitation (CCPR) providing valuable perfusion to end organs. Observational studies have shown an association with the use of ECPR and improved survivability. Two recent randomized controlled studies have demonstrated improved survival with functional neurologic recovery when compared to CCPR. Substantial resources and coordination amongst different specialties and departments are crucial for the successful implementation of ECPR. Standardized protocols, simulation based training, and constant communication are invaluable to the sustainability of a program. Currently there is no standardized protocol for the post-cannulation management of these ECPR patients and, ideally, upcoming studies should aim to evaluate these protocols.

## 1. Introduction

High-performance conventional cardiopulmonary resuscitation (CCPR) only provides about one-third of normal cardiac output. This is not adequate for vital organ perfusion and the resultant low-flow state results in progressive oxygen debt until the return of spontaneous circulation (ROSC) is achieved. Extracorporeal cardiopulmonary resuscitation (ECPR) transitions CCPR to veno-arterial extracorporeal membrane oxygenation (VA ECMO). ECPR effectively ends the low-flow state by reestablishing normal cardiac output and oxygen delivery via extracorporeal support, termed the “return of extracorporeal circulation” (ROEC). ECPR provides time to diagnose, then intervene upon, culprit etiologies without accumulating further oxygen debt and exacerbating ischemia and reperfusion injury. Accordingly, earlier institution has been correlated to neurologically-intact survival. Early, effective, and appropriate ECPR initiation must be paired with effective post-cannulation diagnostics and interventions, alongside high-quality critical care in order to improve outcomes including neurologically-intact survival. This mandates highly coordinated care and effective resource utilization with extreme collaboration between multidisciplinary stakeholders. This manuscript will review potential benefits of ECPR on survival and neurologic outcomes, articulate the necessary requirements for building an ECPR program, and briefly review management considerations after cannulation.

## 2. Cardiac Arrest

Cardiac arrest is an endpoint to a wide variety of conditions. These, plus the duration of the low-flow state, the effectiveness and duration of CPR, and the initial and ongoing clinical care greatly impact patient outcomes. Due to marked differences in etiologies and systems of care, including time-to definitive management, the outcomes after out-of-hospital cardiac arrest (OHCA) and in-hospital cardiac arrest (IHCA) are reported and considered separately. In the US, OHCA has an annual incidence of 74.3 per 100,000 individuals, with approximately 347,000 events occurring annually. Meanwhile, approximately 290,000 IHCA events occur annually in the US [1].

Unfortunately, survival after arrest remains poor [1]. For OHCA, the overall ROSC occurs in 29.7%, with 8.8% surviving to discharge. This is time-dependent, with a probably of “near immediate” ROSC of 60%, decreasing to <10% after 40 min of CCPR [1]. In contrast, for IHCA, ROSC occurs in 48.5% with 25% surviving to discharge [1].

Beyond survival, neurologic outcomes are of utmost importance. During CCPR, low cardiac output will result in irrecoverable cerebral ischemia in every patient at some time-dependent threshold, sometimes as low as 15 min [2]. Typically, post-arrest neurologic outcomes are assessed via short-term functional assessments including the Cerebral Performance Category (CPC) and/or Modified Rankin scale (mRS), with neurologically-intact outcomes defined as a CPC of 1, 2 and an mRS of 0–3 [3]. The international Liaison Committee on Resuscitation also recommends using quality of life tools as additional outcome measures [4]. As expected, neurologic outcomes after arrest worsen in a time-dependent manner. The probability of an eventual mRS score of 0–3 for a patient who suffered OHCA that was treated with CCPR decreases from 30%, if ROSC is achieved within 30 min, to 10% thereafter [1]. More specifically, over 90% of patients who achieve an eventual mRS of 0–3 will achieve ROSC within 20-min of cardiac arrest; all neurologically-intact survivors that are treated with CCPR will achieve ROSC within 40-min. Interestingly, the duration of CCPR for an IHCA does not have clear inflection points that demonstrate decreased neurologically-intact survival. However, similar decrements in outcome would be expected with increased CCPR times [2].

The most favorable clinical features for both OHCA or IHCA include witnessed arrest, an initial shockable rhythm, and early bystander CCPR. These also impact outcomes: an unwitnessed arrest without early bystander CPR has less than a 5% probability of a good neurological outcome [1].

One core contributor to neurologic impairment after cardiac arrest is ischemia and reperfusion injury (IRI), a state of severe oxidative stress, inflammation, and mitochondrial dysfunction that follows ischemia (7). Monitoring and measuring IRI, however, is quite difficult and once ischemia develops, no therapy has shown clear benefit in its management. Measuring for end-organ ischemia is also fraught with difficulty but may offer information that could mitigate IRI severity. Near-infrared spectroscopy (NIRS) reports local capillary-bed hemoglobin oxygen saturation (rSO_2_) and may identify local organ ischemia quickly, enabling goal-directed intervention [5]. Placed over the brain, for example, NIRS can effectively identify cerebral oxygen supply/demand mismatch [5]. In cardiac arrest, NIRS-reported rSO_2_ has been positively associated with both ROSC and functional status at hospital discharge [5]. Notably, ROEC via ECPR can rapidly improve rSO_2_. Remarkably, one study reported >20% improvements in rSO_2_ occurring in as little as 2.5-min after initiation of ECPR [6]. Similarly, animal models suggest inflammatory byproducts of reperfusion injury are also reduced with ECPR compared to conventional CPR [7].

## 3. ECPR for In-Hospital Cardiac Arrest

The first study comparing ECPR to CCPR was done in 2008 for patients suffering from IHCA and reported remarkable improvements in both in-hospital mortality [8]. Notably, a survival difference was described based on the presenting cardiac rhythm. ECPR improved mortality in patients presenting with ventricular tachycardia/fibrillation, but not pulseless electrical activity.

Since this initial study, several small observational studies and associated meta-analyses have been reported comparing CCPR to ECPR for IHCA, yet outcomes are inconsistent [9,10,11]. Nonetheless, data suggested the potential for favorable neurologic outcomes and survival for ECPR versus CCPR for IHCA in patients presenting with a shockable rhythm, when ECPR was initiated within 28 min of arrest, with lower post-arrest lactate levels and with primary cardiac etiologies of arrest [3,4].

As additional observational trials of ECPR accumulated, it became apparent that the time to return of extracorporeal circulation (ROEC) was highly correlated with survival and neurologic outcomes. For time-to-ROEC over 60 min, the rates for survival and neurologically-intact survival both drop to approximately those of CCPR alone [12,13,14,15]. Prior to this, meaningful clinical benefits have been observed [16,17,18,19].

## 4. ECPR for Out-of-Hospital Cardiac Arrest

The ARREST trial was the first randomized, controlled trial that directly compared ECPR to CCPR for OHCA. Limiting enrollment to patients that were presenting with refractory ventricular fibrillation, ECPR yielded a 43% survival to discharge compared to CCPR of <5% (*p* < 0.001), prompting early cessation of the trial by the data safety monitoring board given the apparent benefit [20]. More recently, ECPR was compared to CCPR for witnessed, refractory OHCA (i.e., prolonged cardiac arrest and cardiac arrest without ROSC in the field) in a single-center trial that reported a nonsignificant trend towards survival with good neurologic outcome at 180 days (n = 256, 31.5% vs. 22%, *p* = 0.09) [21]. Importantly, while both of these randomized and controlled trials were stopped early, they both demonstrated the superiority of ECPR not only for survival but also for survival with a favorable neurologic outcome.

## 5. ECPR Program Development

Multidisciplinary effort and well-defined roles and responsibilities are critical throughout the cardiac arrest care continuum; these are emphasized with ECPR. Arrest management includes a wide range of physical locations including the pre-hospital setting, emergency department, cardiac catheterization lab, and intensive care unit. Roles and responsibilities must transition seamlessly, and communication be effective between clinicians in emergency medical services, emergency medicine, cardiothoracic surgery, critical care, perfusion, cardiology, anesthesiology, respiratory therapy, and innumerous additional consultants [22].

### 5.1. Community Approach to OHCA

Centers without expertise and experience initiating and managing VA ECMO are generally poorly positioned to implement a de novo ECPR program. The exception would be a hospital with a close relationship and geographic proximity to an existing ECMO center which could feasibly initiate VA-ECMO for candidate patients, stabilize hemodynamics, and then provide expedited transfer for further management [23].

Many high-functioning EMS systems have adopted a “stay and play” approach to OHCA; that is, prioritizing on-scene ROSC over expedited hospital transport. The underlying philosophy is that high-quality CCPR can be performed in the field, and that intra-arrest patient transport to a hospital that will largely continue similar resuscitative efforts offers little benefit and may be potentially harmful by reducing CCPR quality during transport. High-speed “lights and sirens” ambulance transport is also potentially injurious to EMS providers and the lay public [24].

An ECPR program changes this traditional paradigm since expedited transport facilitates earlier ECPR which in turn improves neurologically intact survival for OHCA. Early collaboration with EMS medical directors to modify existing local protocols is, therefore, essential. Likewise, systematic engagement and education efforts for prehospital providers is critical to improve compliance and shift culture. Emphasis on early consideration of ECPR candidacy is crucial to expedite transport for patients failing to achieve rapid field ROSC (i.e., >5 min of CPR). Regional hospitals in the same EMS catchment should be liaised with to ensure the preferential transport of ECPR candidates to the ECPR center. Given the low survivability of OHCA with CCPR alone, even in the competitive healthcare environment, the authors have found that even local “competitors” are supportive of these initiatives.

### 5.2. Simulation

Simulation can expose weak points and improve planned protocols without risking patient harm. As with any infrequently performed, time-sensitive, and high-acuity procedure, high-fidelity ECPR simulation is critical to mission success [25]. This ideally incorporates the prehospital setting and providers, mimicking an actual emergency medical services call, response, scene resuscitation, and patient transport. The entire initial in-hospital resuscitation and cannulation must be rehearsed. The authors strongly recommend extending the simulation to include transportation and handoff to the cardiac catheterization lab and/or the intensive care unit, as several nonintuitive near-misses can be exposed this way, such as over-small elevators for the expansive equipment and the personnel that are needed. Periodic simulations are encouraged, particularly for lower-volume centers and with ECPR protocol changes.

## 6. ECPR Program Deployment

ECPR is an extraordinarily resource intensive venture for all parties. Early failures can lead to nihilism and even resentment towards the program [26]. Thus, exceedingly conservative inclusion and exclusion criteria should be applied in the early rollout of an ECPR, so as to increase the probability of “early wins” and resultant stakeholder and institutional support.

### 6.1. Prehospital Candidate Patient Identification and Transport

As described, a time-to-ROEC of ≤60 min is highly correlated with neuro-intact survival for ECPR [17]. The challenge is to balance the chances of (preferred) ROSC via CCPR with opportunity to achieve time-to-ROEC <60 min for patients with refractory OHCA. Multiple studies have demonstrated that transport by 16-min after the initialization of CCPR balances the likelihood of on-scene ROSC with ability to achieve the desired ROEC for patients refractory to CCPR [27,28,29]. Simple, memorable adages such as “consider ECPR with the first dose of epinephrine” have been enormously helpful in our experience. 

### 6.2. Hospital-Based Consult and Activation Process

Initial ECPR consultation and activation is critical to minimize low-flow time. The goal is to provide a maximally rapid and sensitive ECPR consultation followed by a specific, robust, and expeditious ECPR activation. Preventable false activations can erode necessary multi-disciplinary stakeholder commitment and response time and should be minimized. Electronic medical record (EMR)-based tools have been described which expedite the process and minimize false activations [30]. Parameters that were included in this navigator encompass the patient age, weight, and arrival room number in the emergency department.

### 6.3. Mechanical Compression Devices

Neurologically-intact outcomes and complication rates are similar for manual and mechanical chest compressions during CCPR [31]. In most functional ECPR programs, if an OHCA victim is deemed to be eligible for ECPR, EMS applies a mechanical CPR device, such as the LUCAS^TM^, (JOLIFE AB, Lund, Sweden) and transports immediately to the ECPR center [32]. To limit personnel in the resuscitation bay and thus maximize access to the patient, mechanical compressions are typically continued during cannulation [17]. Careful attention to the proper placement of the mechanical compression device is crucial in order to maximize the effectiveness of the compressions and to reduce the incidence of compression-related injury in this soon-to-be systemically anticoagulated patient.

### 6.4. Hospital Arrival

For in-hospital ECPR cannulations, teams will realistically have less than 30-min to achieve the 60-min time-to-ROEC goal. The resuscitation and cannulation, including the placement of equipment and people must be scripted and practiced in order to be fast and efficient. Several institutional guidelines have been described and can be tailored to fit a specific system’s need [17,23,33,34]. Consensus guidelines for critical components also exist [35]. A dedicated EMR-based ECPR order set should be released with, or prior to, patient arrival and include necessary medications, laboratory studies, imaging, electrocardiograms, and ECMO parameters.

### 6.5. Equipment and Cannulation

A dedicated closed, wet-primed ECMO circuit should be stored in proximity to whatever clinical arena the ECPR patient is to be cannulated in. Likewise, an ECPR “cart” or “bin” which includes all the necessary cannulation equipment should be maintained adjacent to the circuit. Immediate restocking of both needs to occur following an ECPR activation.

Technique, setting, and the primary operator for cannulation all should be tailored to geography, resources, and institution. Critical action checklists have been described [36]. Both in-hospital and prehospital approaches are in practice, the former being more widely leveraged due to cost and centralization of resources while the latter may confer reduced time-to-ROEC [13,37,38,39]. A percutaneous femoral approach is the most practical, is associated with fewer complications compared to other approaches, and can be performed with ongoing chest compressions [40]. For same-sided cannulation, the femoral artery should be accessed initially to avoid obscuring the smaller target vessel. Surgical cannulation (i.e., femoral cutdown) may be necessary for challenging anatomy, younger patients, or failed percutaneous access [41]. The group of institutional “cannulators” should be large enough to feasibly provide year-round “24/7” coverage, though small enough that every individual in the group performs sufficient cannulations to become and remain competent.

### 6.6. Ongoing Resuscitation

Ideally, the resuscitation leader concentrates on high quality advanced cardiac life support (ACLS) with the goal of ROSC, while the cannulating team works to achieve ROEC. Pulse/rhythm checks are at the discretion of the resuscitation leader, although pauses to compressions should be minimized, and ROSC can be monitored via other endpoints (e.g., with a sudden rise in P_ET_CO_2_). As candidate patients are, by definition, in refractory cardiac arrest, most ECPR centers forgo additional attempts at defibrillation, even if a shockable rhythm is present. This reduces the potential for injury to providers, improves chest compression ratio, and reduces time-to-ROEC. Anecdotally, restoration of cardiac perfusion via several minutes of goal-flow ECMO improves the chances of successful defibrillation (or may convert a non-shockable to a shockable rhythm). Efforts to limit epinephrine boluses toward the end of cannulation can help prevent significant, potentially risky hypertension once the patient is on ECMO support.

### 6.7. Immediate Post-Cannulation Diagnostics and Therapeutics

Post-cannulation care should be part of a protocol, fairly automated, and incorporate the ELSO interim consensus statement guidelines that were published in 2021 [23]. We recommend a checklist approach, with the checklist run immediately after goal flow on ECMO is achieved (Figure 1).

Once a minimal ECMO flow has been attained (e.g., ≥3 L/min), chest compressions are discontinued. Hemodynamics are further stabilized with fluid boluses as well as vasoactive infusions titrated toward a goal MAP of ≥65 mmHg. A right radial arterial line needs to be placed for real-time blood pressure monitoring and for the estimation of cerebral oxygen delivery, especially when native cardiac activity is present or as it recovers later [42].

Initial ECMO settings can vary. Typically, the fraction of delivered oxygen (F_D_O_2_) is set to 100% with sweep gas flow set approximately 1:1 to ECMO flow (e.g., 4 L/min flow would be matched with a sweep flow of 4 L/min) [43]. To ensure durable fever prophylaxis or targeted temperature management (TTM) the heater/cooler is set to 36 °C [44]. The primary management of oxygenation and ventilation should be via the ECMO circuit while the ventilator is leveraged to ensure lung protection, avoidance of ventilator-induced lung injury, and to prevent alveolar decruitment. Typical “rest settings” vary by institution, but reasonable initial settings would include fraction of inspired oxygen (F_I_O_2_) 80–100%, positive end-expiratory pressure (PEEP) 10 cmH_2_O, driving pressure <15 cmH_2_O, and respiratory rate (RR) 8–15 breaths/min [45,46,47]. As cardiac function recovers and blood flow increases through the lungs, the work of oxygenation and ventilation should be transitioned away from the ECMO circuit and towards mechanical ventilation.

A 12-lead electrocardiogram (ECG) is obtained immediately after chest compressions are discontinued. As is typical for post-ROSC ECG’s, the initial post-ROEC ECG interpretability is often limited by the recent global ischemic insult as well as metabolic derangements (e.g., severe acidosis and other electrolyte abnormalities) which often require several minutes to correct. If an ST-elevation myocardial infarction (STEMI) is apparent, particularly in a regional distribution, the patient is taken to the cardiac catheterization lab with the interventional cardiology team. If no STEMI is present, the ECG leads remain in place and a repeat ECG is repeated after 10 min of goal ECMO flow. This typically results in a more reliable and interpretable tracing and may improve capture of intervenable cardiac events.

Point-of-care ultrasound (POCUS) can provide substantial actionable information [48]. We evaluate for the presence of pericardial effusion, assess the global biventricular size and function, and, if there is organized cardiac activity, determine whether aortic valve opening is present. The ECMO cannulas are next interrogated; specifically, the distal tip of the drainage cannula should be at the level of the right atrium. The lung fields are evaluated to evaluate for the presence of pneumothorax, consolidation, edema, or pleural effusions. Finally, the abdomen/pelvis is viewed via a typical focused assessment with sonography in trauma (FAST) windows to assess for obvious free fluid an indicative of injury related to CCPR. 

Most institutions have imaging protocols including head, chest, abdomen, and pelvis cross-sectional tomography for all ECPR patients, even if this means a short delay in time to definitive intervention such as catheterization lab. Imaging may reveal catastrophic CNS injury, help elucidate the etiology of OHCA, and diagnose clinically important hemorrhage secondary to prolonged chest compressions. Following imaging, ideally the patient is transported directly from to the next phase of care, such as the cardiac catheterization lab or an intensive care unit (ICU).

A multidisciplinary discussion should quickly occur to determine the need for a distal perfusion catheter for the lower extremity, and to consider left ventricular (LV) venting. Considerations for these strategies will be discussed below.

### 6.8. Cardiac Catheterization Lab

Complications of coronary artery disease are major etiologies of cardiac arrest [16]. As detailed above, we recommend that ECPR be paired with robust and facile access to coronary angiography and associated percutaneous coronary intervention, for ready activation, particularly in the absence of a clear alternative non-cardiac cause [17,20,23,32].

Immediate percutaneous coronary intervention has been associated with positive post-arrest outcomes in several single center studies, even in the absence of ST-segment elevation, including duration of ECMO, neurologic outcomes and mortality [45,49,50]. In contrast, two recent randomized trials examining patients with ROSC after non-refractory OHCA (i.e., not requiring ECPR) did not support benefit of early versus delayed/selective cardiac angiography for those without ST-segment elevation MI [51,52].

Nonetheless, in addition to diagnosing and intervening upon acute coronary syndromes, early catheterization allows for additional diagnostics (e.g., pulmonary angiography) and therapeutics (e.g., placement of a distal lower extremity perfusion catheter or percutaneous left ventricular vent), which may offer significant patient benefit.

## 7. ECPR Quality Improvement and Patient Safety (QI/PS)

### 7.1. ECPR Registry Maintenance and Participation

An EMR-based consult and activation tool enables ease of ECPR registry generation. In addition to all formal ECPR consults, the authors recommend routine interrogation of all regional OHCA calls in the program catchment to assess for ECPR candidacy. Feedback and trends should be shared with prehospital and emergency department providers to improve the sensitivity and specificity of the consult and activation process. Similarly, institutions with an ECPR program are encouraged to leverage the international Extracorporeal Life Support Organization (ELSO) registry.

### 7.2. Geospatial Mapping

Our center leverages a commercially available geospatial mapping tool ArcGIS^TM^ (Ersi, Redlands, CA) to generate a county-wide map displaying anticipated time-to-ROEC based on arrest location. Local traffic patterns, historical EMS transport times, duration of on-scene resuscitation, and ED-to-cannulation time are integrated into the mapping tool and can be independently modified. Approximate coordinates of ECPR scene calls, responding EMS agency, and patient outcome can be superimposed on this map. The location of arrest, combined with these modifiers, can thus objectively be incorporated into patient candidacy (Figure 2).

The outcomes for OHCA patients with anticipated time-to-ROEC <60 min should be monitored for actionable trends using this tool [53]. For example, at our institution, geospatial mapping identified a cluster of cases close to our hospital with poorer than expected outcome, prompting additional analysis. This demonstrated that transport time was likely less impactful on outcomes than limiting on-scene resuscitation time for patients that were accepted for ECPR.

### 7.3. Multidisciplinary QI

Every single ECPR activation should reveal opportunities for improvement. Timely, focused debriefs by program leadership and participating prehospital personnel, resuscitationists, and other clinicians are invaluable. Periodic formal multidisciplinary stakeholder case reviews as well as protocol modifications are equally essential.

## 8. Post-Cannulation ICU Management

It is clear that post-cardiac arrest care is a critical part of the chain of survival. There is increasing interest in identifying and optimizing practices that are likely to improve outcomes. After initial stabilization, post-arrest critical care includes targeted hemodynamic support, optimized mechanical ventilation, goal-directed temperature management, diagnosis and treatment of underlying conditions and causes of arrest, early identification and treatment for seizures, infections, and management of organ dysfunction and failure. As many patients who survive the initial arrest will die due to withdrawal of life-sustaining treatment in the setting of neurological injury, much of post-arrest care focuses on decreasing injury to the brain [54]. Goal-directed end-points include optimized cerebral perfusion pressure, maintenance of normoxia and normocarbia, targeted temperature management and/or avoidance of fever, detection and treatment of seizures, and more [55,56].

### 8.1. Coronary Imaging and Intervention

As discussed above, coronary angiography should be considered for cases that are thought to originate from a cardiac nature. Not unexpectedly, initial left ventricular ejection fraction is often severely reduced on the initial echocardiogram. However, after 72 h the systolic function recovers in most patients, suggesting that serial echocardiographic examinations should be performed on all ECPR patients [20].

### 8.2. Temperature

The AHA guidelines recommend targeted temperature management (32–36 °C) for at least 24 h after achieving the target temperature for all cardiac rhythms in both OHCA and IHCA [56]. It remains unclear exactly what temperature target or strategy optimizes neurologic outcomes [57]. While lower temperatures may offer direct neuronal benefit, these are juxtaposed with complications of hypothermia, including coagulopathy, dysrhythmias, electrolyte disturbances, and more.

At present, there is no RCT data published for patients undergoing ECPR; therefore, we must infer from other data to guide practice. A large randomized and controlled trial recently demonstrated that a targeted temperature of 33 °C did not confer benefit when compared to a targeted temperature of 36 °C [44]. This was followed by a subsequent negative trial, with no difference in all-cause mortality or functional outcomes when 33 °C was compared to maintenance of normothermia (37.5 °C) [56]. More recently, favorable neurologic outcomes at 90 days were demonstrated when 33 °C was targeted in a specific cohort of OHCA-non-shockable arrests [58]. This trial was than replicated for patients that were suffering from IHCA with a target temperature of 32–34 °C. Unfortunately, it was stopped early due to futility, noting problematic methods and inconsistent fever management [59].

TTM is associated with an increased risk of secondary infections [60,61] and is an independent risk factor for early ventilator-associated pneumonia (VAP) [60,62]. A recent RCT investigated the rate of VAP in patients that suffered a cardiac arrest with a shockable rhythm who were subsequently cooled. They demonstrated a reduction in the rate of VAP in patients that were treated with a short course of antibiotic therapy at seven days [63]. The cohort was not placed on ECPR, however, the data may be extrapolated to those ECPR patients that undergo TTM.

### 8.3. Blood Pressure

Hypotension may worsen cerebral and other end-organ injury, however, the optimal mean arterial pressure (MAP) after ROSC remains unclear. Generally, a MAP of 63–83 mmHg has been accepted. Increasing the MAP further has not been shown to improve survival and may decrease ECMO flow due to afterload [64,65].

### 8.4. Anticoagulation

Systemic anticoagulation is used to prevent thrombotic complications from the patient-circuit interface but increases the risk of gastrointestinal and intracranial bleeding. The risks and benefits of systemic anticoagulation must be re-assessed frequently; the most common complication of ECMO is bleeding. The cannulation site as the most common site of bleeding [55,66].

Although systemic anticoagulation is needed to maintain efficacy of the circuit, coagulation status must be frequently assessed and the risk of bleeding with thrombosis needs to be balanced. ECPR patients are at increased risk for bleeding due to antiplatelet administration (if PCI was performed), hypothermia, acidosis, and coagulopathies such as DIC, platelet dysfunction, thrombocytopenia, and hemodilution. DIC, specifically, occurs with alarming frequency in this population, with approximately half of OHCA patients and one-third of IHCA patients in one series [67]. It may not be reasonable to apply the same anticoagulation protocols as for conventional VA-ECMO recognizing that ECPR patients may require repeated neuroimaging [67,68].

### 8.5. Left Ventricular Venting

Left ventricular venting should be considered when there is evidence of distended LV with limited unloading. This may be illustrated by echocardiography with the demonstration of LV distention, left atrial distention, decreased frequency of aortic valve opening, or smoke in the cardiac chambers. We can also observe the “failure to empty LV” signs in arterial pressure waveform tracing by a loss of pulsatility, decreased pulse pressure (<10 mmHg), or increasing filling pressures (pulmonary capillary wedge pressure > 25 mmHg, central venous pressure A > 20 mmHg) despite acceptable ECMO flows (>3 L/min) [47]. If the patient is already destined for immediate cardiac catheterization, LV venting interventions (e.g., Impella placement, IABP placement, atrial septostomy) can be performed following coronary angiogram and percutaneous coronary intervention (PCI).

### 8.6. Limb Ischemia

The arterial cannula can obstruct blood flow to the limb with femoral cannulation for ECMO in a cannula- and patient-size-dependent manner. This may cause limb ischemia that is severe enough to require fasciotomies or even amputation. The reported incidence of limb ischemia complicating VA-ECMO ranges from 10 to 70% [69]. A distal perfusion cannula may help decrease this complication. Unfortunately, the risk is not eliminated and 12–33% of patients develop limb ischemia despite its use [70,71].

The use of smaller arterial cannulas (15–17 Fr) may preempt the need for distal limb perfusion. Individualized decision-making can be informed by limb near-infrared spectroscopy (NIRS). Tissue saturation by NIRS should be above 50%, ideally >60%, and there should be less than a 20% saturation difference between the two extremities [72]. If tissue ischemia is suspected, a distal perfusion cannula may be placed in the superficial femoral or posterior tibial arteries [72].

### 8.7. Prognostication of Neurologic Recovery

By the nature of ECMO’s ability to provide essentially normal cardiopulmonary perfusion, withdrawal of life-sustaining therapy (WLST) is one of few means where patients on ECMO support can actually expire. Guidelines for neuroprognostication of ECPR patients are largely extrapolated from CCPR patients. These suggest avoiding early WLST within the first 72 h post initial arrest (or post-rewarming for comatose patients) [73,74]. Characteristic features on serial electroencephalograms (EEG) have been shown to correlate with neurologic outcome and when combined into the new cerebral recovery index may aid the WLST decision [75]. To date, no studies have attempted to determine an appropriate duration of VA ECMO prior to WLST in ECPR patients.

## 9. Summary

It is increasingly clear that ECPR is superior to CCPR for neurologically-intact survival after cardiac arrest when leveraged for appropriate patients, paired with outstanding CCPR, and when instituted quickly. It is best implemented at high volume centers, with organized protocols, coordination amongst specialties, and with excellent post-arrest care. Incredible multidisciplinary buy-in and support are necessary to build and maintain an ECPR program. Attentive goal-directed post-cardiac arrest care and avoidance of early withdrawal of life-sustaining therapy are key to maximizing the benefits of ECPR.

## Figures and Tables

**Figure 1 medicina-58-01815-f001:**
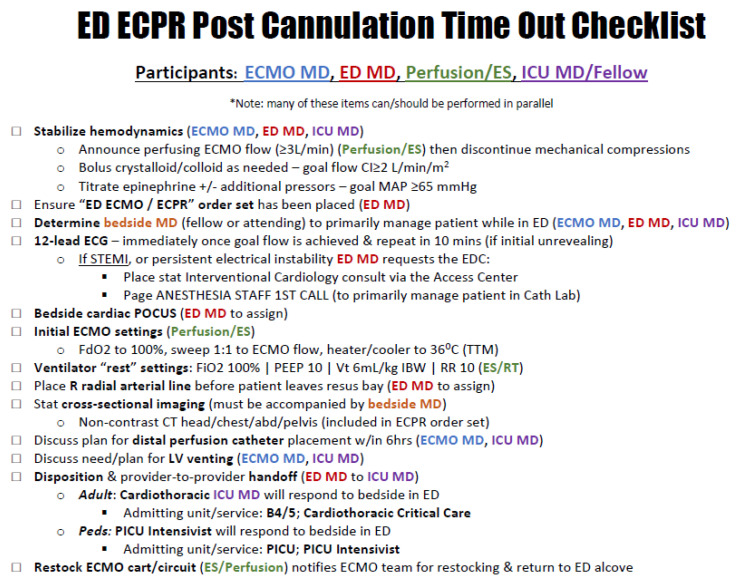
An example of a post-ECPR-cannulation checklist. ECMO MD = cannulating physician; ED MD = emergency medicine physician; perfusion/ES = perfusionist or ECMO specialist; ICU MD/Fellow = critical care physician. CI = cardiac index. MAP = mean arterial pressure. STEMI = ST segment elevation myocardial infarction. Access Center = centralized paging and operator resource for the hospital system. POCUS = point-of-care ultrasound. FdO2 = fraction of delivered oxygen. LV = left ventricular. B4/5 = institutional cardiothoracic ICU (or wherever adult VA ECMO patients are managed). PICU = pediatric intensive care unit.

**Figure 2 medicina-58-01815-f002:**
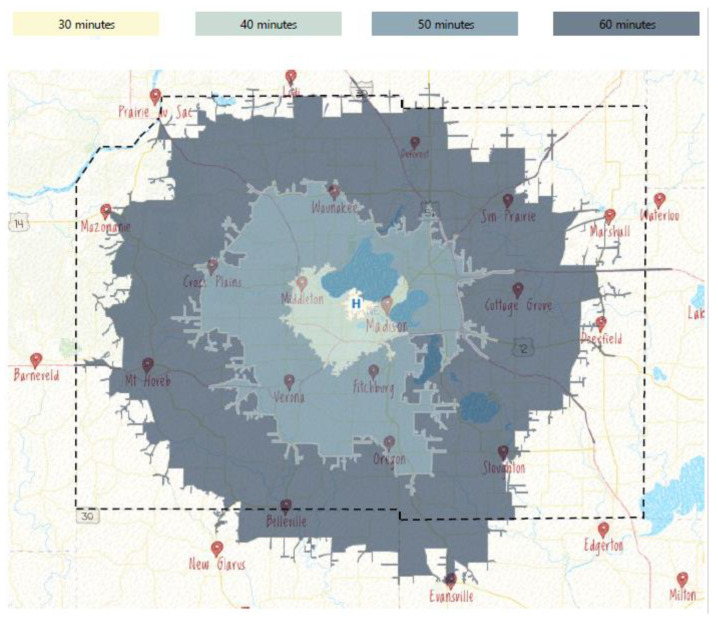
Projected time from witnessed arrest to ECMO cannulation. This map describes time-to-ROEC for patients, assuming a 2-min delay in calling 911. It integrates historic scene–response time based on geographic location (typically <5 min county-wide), a 16-min on-scene resuscitation by paramedics, rapid transport back to the Emergency Department (drive-times are based on Wednesday at 0830, the 75th percentile for this metric), and a 15 min door-to-cannulation time. The latter is ambitious, but in line with our simulated and historic times. Color-coded boxes reflect the minimum projected time-to-ROEC. Accordingly, the outer “60 min” area is considered feasible only under ideal circumstances and for excellent ECPR candidates.

## Data Availability

Geospatial mapping may be found at: https://www.arcgis.com/home/index.html.

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
