# Peer review of "Extracorporeal Cardiopulmonary Resuscitation: A Narrative Review and Establishment of a Sustainable Program"

_medicina, 2022, doi:10.3390/medicina58121815_

Round 1

Reviewer 1 Report

In their paper, Cassara et al summarized the available evidence regarding the use of ECPR for refractory cardiac arrest. The recent evidence establishing the survival benefit of ECPR makes the review clinically relevant and a useful resource for clinicians. However, the following issues need to be addressed:

1) There needs to be a critical separation of the evidence stemming from refractory and non-refractory cardiac arrest. For example, the patients mention the COACT trial in the Cardiac Catheterization Lab section. The COACT trial had enrolled exclusively non-ecpr patients and whether its results can be applied to refractory cardiac arrest is not known.

2) Please include a reference for Figure 3

3) Please provide clarifications regarding some terms in Figure 1. What does B4/5 mean?

4) Extensive grammar and vocabulary editing would be needed before resubmitting

5) On the section regarding temperature management please cite the most recent RCTs, TTM-2 and the HACA-IHCA trial.

6) While I commend the authors for their work, I want to recognize that ECPR is a vast topic with multiple parameters extending from field resuscitation to post-cardiac arrest care and neurocritical care. Attempting to condense all this information to the limited space a review provides might result in the loss of valuable information that might make comprehension of the article difficult. I would recommend removing the post-cannulation part so that the rest of the review, especially the geospatial mapping which is the most important piece of innovation in this article, can be adequately covered.

Author Response

Reviewer 3

C1: This a very well written review on establishing an eCPR program. Appropriate literature review is included and many different aspects that need to be considered when initiating an eCPR program are noted.  Manuscripts such as this are very important as many centers explore the feasibility. I think missing in the manuscript is a more in-depth discussion of the challenges of development and implementation needs to be included.  As noted, by the authors "incredible multidisciplinary buy in is needed".  This is true both of the institution developing the eCPR program but also for the necessary partnerships with other entities that will participate in some aspect of the patient's care. There needs to be both institutional infrastructure that can meet the skill, cost and care needs of these patients and a regional infrastructure that can facilitate such a time sensitive process and also support an eCPR patient appropriately after cannulation. The many hurdles associated with an eCPR program are insurmountable at this time by many institutions. This should be delved into a little more as many of the hurdles aren't readily apparent but would be useful to learn from literature such as this article.

R1: We appreciate this reviewer’s keen eye and prospective about the hurdles in starting an ECPR program.  In our review, we have attempted to help the readers anticipate and mitigate the challenges associated with the beginning of an ECPR program. While there are certainly a multitude of impediments to this, mostly cost and resources, we hope that our manuscript can explain the workable solutions that our institution has implemented.  We have expanded our deployment section including geospatial mapping.

Reviewer 2 Report

I reviwed to Cassara CM et al's manuscript as "Extracorporeal cardiopulmonary resuscitation: A narrative reiview and establishment of sustainable program" for publication possibility in Medicina. 

This review manuscript is well-planned and written about the ECPR. I think that it should be accepted with this form.

Best regards

Author Response

We thank the reviewer for their scrutinization of our manuscript and thank them for the kind words. 

Reviewer 3 Report

This a very well written review on establishing an eCPR program. Appropriate literature review is included and many different aspects that need to be considered when initiating an eCPR program are noted.  Manuscripts such as this are very important as many centers explore the feasibility. I think missing in the manuscript is a more in-depth discussion of the challenges of development and implementation needs to be included.  As noted, by the authors "incredible multidisciplinary buy in is needed".  This is true both of the institution developing the eCPR program but also for the necessary partnerships with other entities that will participate in some aspect of the patient's care. There needs to be both institutional infrastructure that can meet the skill, cost and care needs of these patients and a regional infrastructure that can facilitate such a time sensitive process and also support an eCPR patient appropriately after cannulation. The many hurdles associated with an eCPR program are insurmountable at this time by many institutions. This should be delved into a little more as many of the hurdles aren't readily apparent but would be useful to learn from literature such as this article.

Author Response

Reviewer 1:
C1: There needs to be a critical separation of the evidence stemming from refractory and non-refractory cardiac
arrest. For example, the patients mention the COACT trial in the Cardiac Catheterization Lab section. The COACT
trial had enrolled exclusively non-ecpr patients and whether its results can be applied to refractory cardiac arrest is
not known.
R1: We thank the reviewer for this astute observation and agree. We have edited the document in an attempt to
delineate the evidence from refractory and non-refractory cardiac arrest. We have paid special attention to the
“Cardiac Catheterization” section lines 321-339
C2: Please include a reference for Figure 3
R2: The Figure was removed from the document as we do not believe it added to the understanding of the
paragraph. Lines 388-392
C3: Please provide clarifications regarding some terms in Figure 1. What does B4/5 mean?
R3: Figure 1 was moved to the end of the document with an added in text call out. (line 265). All terms were
defined in the explaination of the figure
C4: Extensive grammar and vocabulary editing would be needed before resubmitting
R4: We appreciate the keen eye of this reviewer and agree that the document required a fine-tooth comb prior to
being acceptable for final publication. Accordingly, we completely revisited the grammar and narrative approach
to the document and made extensive changes to wording and sentence structure throughout, with associated
editing. Further, we made some systematic adjustments to the flow, ie. combining the cardiac catheterization
sections. We will supply the edited document with tracked changes if needed.
C5: On the section regarding temperature management please cite the most recent RCTs, TTM-2 and the HACAIHCA trial
R5: We thank the reviewer for bringing these 2 trials to our attention. Both have been cited and analyzed in the
TTM section. Lines 391-415
C6: While I commend the authors for their work, I want to recognize that ECPR is a vast topic with multiple
parameters extending from field resuscitation to post-cardiac arrest care and neurocritical care. Attempting to
condense all this information to the limited space a review provides might result in the loss of valuable information
that might make comprehension of the article difficult. I would recommend removing the post-cannulation part so
that the rest of the review, especially the geospatial mapping which is the most important piece of innovation in
this article, can be adequately covered.
R6: We value this reviewer’s observation that more time should be spent on the geospatial mapping. We have
expanded the narrative on this section. Unfortunately, to our knowledge, there is little data regarding EMS times
and time-to ROEC therefore we spent time explaining the experience at our own institution. We also believe that
the post cannulation management reviews the most current literature and adds to the knowledge of this subject as
a whole. We would appreciate the Journal’s consideration to leave the text as-is.